# A Stochastic Frontier Model for Definition of Non-Technical Loss Targets

**Daniel Leite [1], José Pessanha [2], Paulo Simões [3], Rodrigo Calili [4],***  **and Reinaldo Souza [4]**

[1]  Enel Brasil, Niterói 24020-005, Brazil; daniel.leite@enel.com
[2]  Institute of Mathematics and Statistics, Rio de Janeiro State University, Rio de Janeiro 20550-000, Brazil; pessanha@ime.uerj.br
[3]  Brazilian Institute of Geography and Statistics, Rio de Janeiro 20021-120, Brazil; paulomahaz@gmail.com
[4]  Department of Industrial Engineering, Pontifical Catholic University of Rio de Janeiro, Rio de Janeiro 22451-900, Brazil; reinaldo@puc-rio.br
*  Correspondence: calili@puc-rio.br

**Abstract:** The theft of electrical energy is one of the main problems faced by electricity distribution utilities, especially in developing countries. Aware of the difficulties in combating non-technical losses (NTLs) in Brazil, the National Electric Energy Agency (ANEEL) established tolerable limits for the percentage of non-technical losses to each Brazilian distribution utility. Despite the notable progress made by ANEEL, when comparing public utility NTLs and their regulatory targets in the last decade, it was observed that the goals defined by this agency were not able to lead to a general reduction in NTLs in the country. Thus, the search for alternative methodologies to deal with the topic is necessary. A more attractive alternative to the ANEEL's model is an efficient frontier model. This paper describes a stochastic frontier cost model for panel data whose equation is specified to provide the tolerable limits for the percentage of NTLs. The proposed model was applied to a panel of data containing annual observations, over 10 years, of 41 distribution utilities in the Brazilian electrical system.

**Keywords:** non-technical losses; distribution; stochastic frontier analysis; panel data

## 1. Introduction

The theft of electric power is one of the main problems faced by distribution utilities in developing countries, in particular in many countries in Africa, Latin America, and South Asia [1–5], where the current levels of the theft of electric power pose risks to not only the solvency of many electricity distribution companies in these countries but also to the security of energy supply itself [1,3,4]. In Brazil, non-technical losses of electricity (NTLs) have remained relatively stable over the last decade despite the efforts of the National Electric Energy Agency (ANEEL) and utilities to combat this problem, as shown in Figure 1 by the percentage of non-technical losses (PNTL) in the total energy injected into the utility grid.

Though the situation in Brazil is much less dramatic than that observed in other developing countries (and, overall, it is even close to the average 7% observed in the OECD countries [4]), NTLs have a significant financial impact both for utilities and other Brazilian consumers. This is due to the fact that in addition to the costs of the generated and unpaid energy itself, consumers and utilities companies have to bear the full cost of the transmission and distribution infrastructure associated with these NTLs.

In Brazil, the NTL is an input variable for tariff calculations. The NTL is partially passed to consumers through electricity tariffs, i.e., the NTL is a cost shared among all consumers [6].

The transference of the NTL to the tariffs is justified by the fact that the NTL depends on factors not manageable by the utilities. However, this solution does not totally solve the problem because it inflates the tariffs, and, consequently, it encourages defaults [7] and more theft of electricity [1]. Nowadays, the non-payment rate is another problem faced by Brazilian distribution utilities [7]. The non-technical losses are hampering efforts to achieve lower tariffs and greater improvements in energy efficiency. Ultimately, non-technical losses deteriorate the economic and financial balance of utilities, and they jeopardize the sustainability of electric power supplies [8].

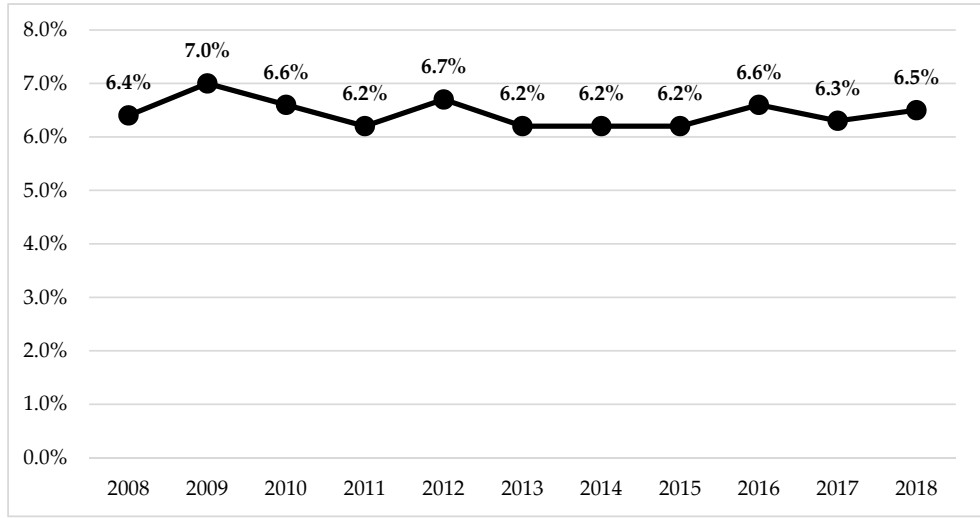

**Figure 1.** Percentage of non-technical losses in the total energy injected into the utility grids [9].

Another aspect that makes the regulatory treatment of NTLs very complex in Brazil is the relative heterogeneity of NTLs observed among electric distribution utilities (Figure 2). In 2018, for example, of the 54 main utilities, 25 of them registered PNTLs below 3% of the total injected energy, values very close to the PNTL values observed in countries such as the US and Canada. On the other hand, 11 of these utilities had PNTLs higher than 10% of the total injected energy [10], reaching, in some cases, more than 30%, which are values comparable to those observed in countries like India and Bangladesh [11].

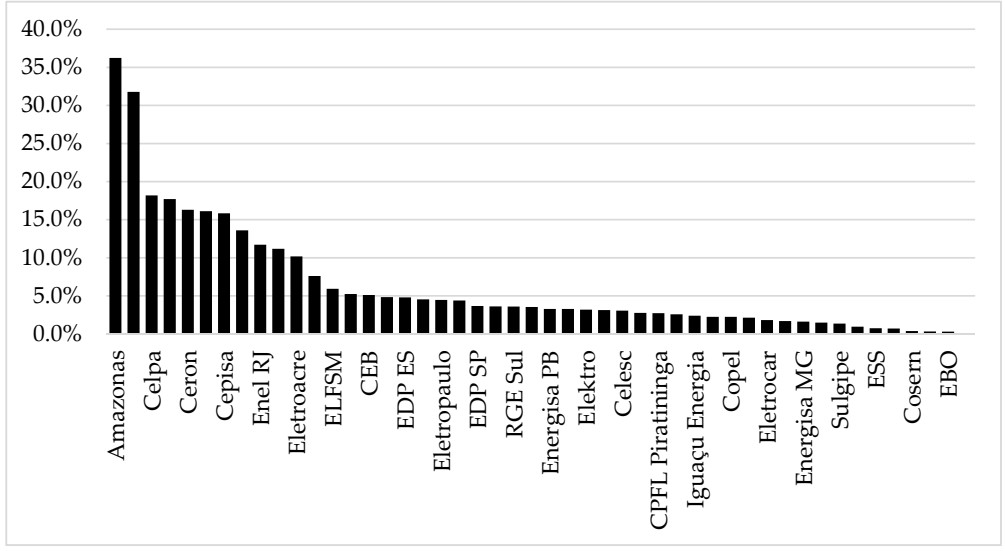

**Figure 2.** Percentages of non-technical losses (PNTLs) in the total energy injected into the utility grids in 2018 by main utilities [9].

In order to reduce the harmful effects of NTLs, the distribution utilities should invest in technology for theft detection and the inspection of consumer units to identify and punish fraudsters [1,8,12–14]. However, the costs are high and they can outweigh the benefits from the non-technical loss reduction if the investments are greater than the energy losses cost reduction [6].

NTLs will never totally be eradicated [4]. There is a limit tolerable to NTLs, from which the costs to reduce them outweigh the benefits of their reduction. It is known that the characteristics of the environment in which the utilities are inserted can dramatically influence their outcomes in the fight against NTLs [4,12]. Experience has shown that combating losses in some areas is much more challenging than in others. The difference is associated with a number of variables, especially those associated with the socioeconomic characteristics of the region, such as the criminality and inequality levels, the infrastructure quality, and the strength of local Institutions [4,5,11,15]. This means that optimal the minimum level of NTLs that needs to be achieved in a more socioeconomic complex area tends to be higher than in others.

On the other hand, the level of effort undertaken by utilities in combating NTLs affects their PNTL levels. Some utilities have achieved substantial reductions in NTLs even under more adverse conditions through technological innovation, management improvements, and investments. As the socioeconomic reality of the concession areas has not substantially changed, it is possible to attribute the observed reductions to a more efficient management in the fight against non-technical losses.

If it were possible, through observation, to decompose the NTL levels of each utility into two installments, one resulting from the actions carried out by the utility (manageable portion) and the other related to the environment in which the utility is inserted (non-manageable portion), the regulator's problem would be to determine the optimum NTL level of the manageable portion by ensuring the full transfer of the non-manageable portion—a process that could be performed quite simply. However, it is not possible to make this decomposition directly and accurately, so the regulator must define a method that allows him/her to separately estimate these parcels.

Aware of this limit and aiming to reduce losses, the ANEEL [10] established a tolerable limit or target value to the PNTL for each utility. Any percentage of non-technical losses above the target implies burdens that must be assumed by the utilities and not by their customers.

The target value depends on the socioeconomic complexity [10] in the concession area, which is a construct based on the following variables: the proportion of subnormal households, garbage collection coverage, income inequality (Gini Index), credit default levels, violent death rates, and the proportion of low-income clients. The level of socioeconomic complexity is achieved by fitting an econometric model for panel data in which the dependent variable is the percentage of non-technical losses and the explanatory variables correspond to the above-mentioned socioeconomic variables [10,16].

Despite the remarkable progress promoted by the ANEEL, Figure 3 shows that the targets defined by the ANEEL have been unable to lead the utilities to a cycle of reductions of general NTLs in the country.

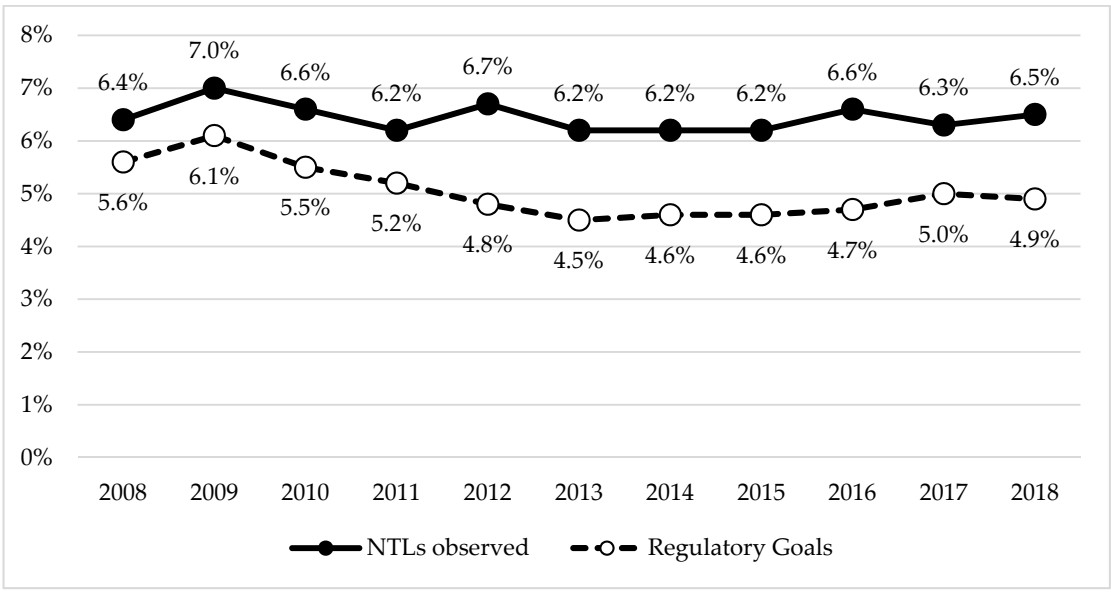

**Figure 3.** PNTL in Brazil vs. regulatory targets (% of total injected energy) [9].

This issue is even more serious when individually analyzing utilities. From 2008 to 2018, according to data from the ANEEL, an average of 68% of the country's 54 main utilities were unable to meet the regulatory loss targets set by the ANEEL. The situation is even more dramatic if we consider only the results of the last three years, during which 77% of utilities have been unable to meet their NTL regulatory targets, as illustrated in Figure 4.

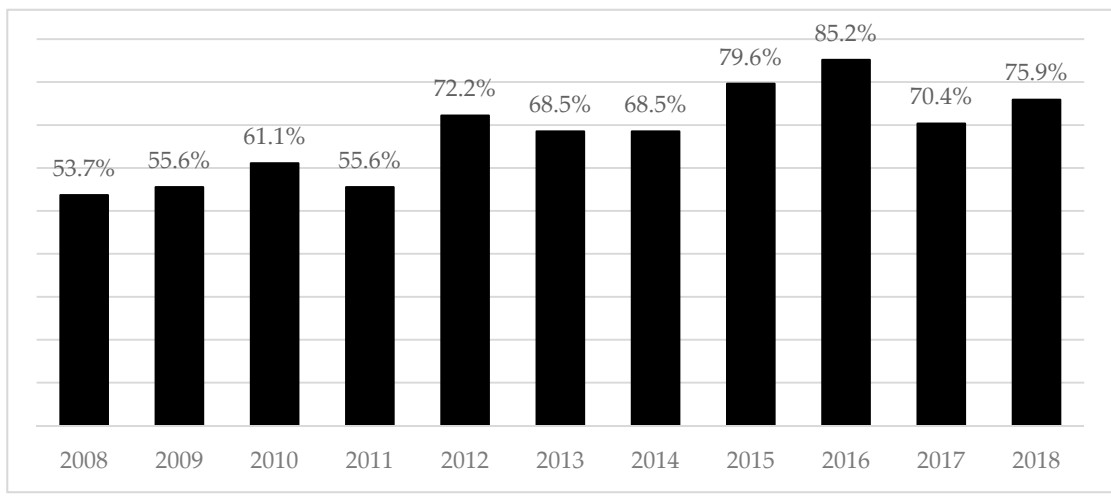

**Figure 4.** Share of distribution utilities that failed to reach their regulatory PNTLs over the period of 2008–2018 [9].

Thus, the search for alternative methodologies to deal with the topic seems to be necessary. A more attractive alternative to the ANEEL's model is an efficient frontier model [17–20]. The efficiency frontier for the PNTL is a benchmark that can be identified through data envelopment analysis (DEA) [21]. An alternative to DEA models is the stochastic frontier analysis (SFA) model [22].

The DEA and SFA approaches aim to estimate an efficiency frontier from data, but they differ in the methods employed; DEA is a non-parametric approach based on linear programming [23], while SFA is a parametric approach that relies on econometric modeling [24]. Additionally, in the DEA approach, the effort undertaken by an utility to reach the benchmark (frontier) corresponds to the utility's deviation from the efficiency frontier. On the other hand, in the SFA approach, there is the recognition

that part of the deviation from the frontier is due to factors that are not manageable by utilities [22,24], a premise that is compatible with the reality faced by utilities in combating non-technical losses.

Aiming to improve the transparency and reproducibility of the regulatory procedures adopted by the ANEEL to control non-technical losses, the present work describes a panel data SFA model to provide the PNTL targets for all Brazilian distribution utilities as an alternative to the econometric approach used by the ANEEL in the last tariff review cycle [16]. The choice of the SFA approach was due to the recognition that non-technical losses are determined by variables that are not manageable by utilities. Since the non-technical losses are costs, the utilities must minimize them. Thus, the SFA model proposed in this work is like a stochastic cost frontier model [22,24]. Recently, a similar approach based on a panel data SFA model was used to evaluate the energy and carbon efficiency for emerging countries [25].

It is worth noting that the DEA and SFA models have been successfully applied in the economic regulation of electricity distribution and transmission utilities, particularly in the definition of the regulatory operational expenditure (OPEX) for each utility, a key element for the annual allowed revenue assessment. [18–20,23,26].

This paper is organized in five sections. Next, Section 2 outlines the basic theoretical stochastic cost frontier model for cross-section and panel data in Sections 2.1 and 2.2, respectively. The specification of the proposed SFA model and the method to compute the PNTL targets are described in Section 2.3. The proposed methodology was applied to panel data containing annual observations over 10 years of 41 distribution utilities in the Brazilian electrical power system and the achieved results are displayed and discussed in Sections 3 and 4, respectively. Finally, Section 5 summarizes the main conclusions.

## 2. Materials and Methods

The cost frontier indicates the minimum cost required to produce a quantity of product given inputs prices and technology. Thus, inefficient producers are located above the frontier, while efficient producers are at the cost frontier, as shown in Figure 5.

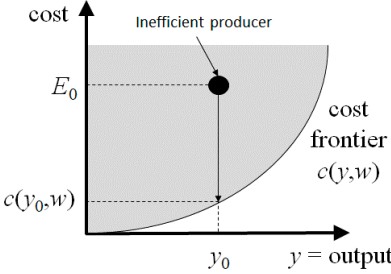

**Figure 5.** Cost frontier.

The cost frontier is a benchmark against which the performance of producers in the same industry sector (decision making units—DMUs) can be compared; in this work, the DMUs are the Brazilian distribution utilities. A comparison with the frontier function allows for the classification of the DMUs into the categories of efficient and inefficient.

It is necessary to recognize that the analyzed problem does not belong to the field of microeconomics. The analyzed variables do not include the variables considered in the theory of production and cost. Thus, the microeconomic theory does not present prescriptions about the relationship between NTLs and their drivers.

However, it is also necessary to recognize, that despite that, the non-technical loss is a variable that must be minimized, which is why the theoretical framework of the cost frontier was adopted.

## 2.1. Cost Stochastic Frontier Model for Cross Section Data

The deviations from the frontier function reflect failures in management optimization. This suggests that the degree of relative efficiency of a DMU can be evaluated by its distance from the frontier, using the radial metric [22,24], which is a number in the interval [0,1]: the DMU is considered efficient if the metric is equal to one; otherwise, it is considered inefficient. For a DMU to produce a quantity y of product from a quantity x of inputs with unit prices w, the efficiency $\theta$ is the ratio of the minimum potential cost defined by the efficiency frontier c (y,w) and the production cost E = wx $\geq$ c (y,w):

$$\theta = c\ (y,w)/E \tag{1}$$

By arranging the terms of Equation (1), E = c (y,w) $\theta^{-1}$. Then, the following equation can be obtained after a natural logarithmic transformation:

$$\log(E) = \log(c\ (y,w)) - \log(\theta) \tag{2}$$

Assuming that the cost function is linear in the natural logarithms of the variables (e.g., a Cobb–Douglas specification) and that $\varepsilon = -\log(\theta)$, which is a random term, we have the following linear regression equation for each DMU i in a set with n DMUs:

$$\log(E_i) = \beta_0 + \beta_1.\log(y_i) + \beta_2.\log(w_i) + \varepsilon_i \ \forall\ I = 1,n \tag{3}$$

In Equation (3), the random term $\varepsilon_i$ expresses the deviation between the verified cost ($\log(E_i)$) and the minimum cost defined by the efficiency frontier ($\beta_0 + \beta_1.\log(y_i) + \beta_2.\log(w_i)$). It is noteworthy that, unlike the conventional linear regression model, the random term in Equation (3) has a non-zero mean ($E(\varepsilon_i) > 0$) and it is not normally distributed [22].

In general, the SFA models are specified as Cobb–Douglas (CD) or Translog (TL) forms [17,24]. In the case of the cost frontier, the Translog cost function has the most favorable functional properties because it is flexible, but this approach also has problems because it is not parsimonious (there are more parameters to estimate), and this may give rise to econometric difficulties such as multicollinearity and the need for larger samples.

In addition, the Translog cost function collapses to a Cobb–Douglas cost function, and the latter is a particular case of the former. The Cobb–Douglas function is less flexible than Translog, but it is parsimonious, i.e., it is the simplest functional form that "gets the job done adequately" [17].

A good example of the application of Cobb–Douglas form is the recent comparative study of energy and carbon efficiency for emerging countries by using panel stochastic frontier analysis [25].

In deterministic frontier models, any deviation from the frontier is attributed to inefficiency. Such models ignore the fact that costs can be affected by random shocks not manageable by the DMUs. One advance in this regard is the stochastic frontier model, whose main virtue lies in the recognition that deviations from the frontier may originate from the inefficiency of the producers or may be caused by unmanageable random shocks. In order to accommodate the two sources of deviations, the SFA decomposes the random term into two components ($\varepsilon_i = v_i + u_i$):

$$\log(E_i) = \beta_0 + \beta_1.\log(y_i) + \beta_2.\log(w_i) + v_i + u_i \ \forall\ I = 1,n \tag{4}$$

In Equation (4), $v_i$ is a normally distributed random component with zero mean that picks random shocks not manageable by the i-th DMU, while $u_i$ is a nonnegative random component that catches the effect of the degree of inefficiency of the i-th DMU. The sum of the random components defines the compound error $\varepsilon_i = v_i + u_i$ as positively asymmetric.

The stochastic frontier has two parts: a deterministic part, common to all DMUs ($\beta_0 + \beta_1.\log(y_i) + \beta_2.\log(w_i)$), and a specific part of each DMU, i.e., the component $v_i$ that captures the effects of random shocks. The efficiency measure of the i-th DMU is given by:

$$\theta_i = \exp(-u_i) \tag{5}$$

Equation (4) is estimated by maximum likelihood, and, given that the random variables u and v are unobservable, there is a need to make some assumptions about their probability distributions. Usually, the more common assumptions are a normal distribution for v and a half-normal distribution for u, a specification known as a normal/half-normal SFA model [26]:

1. $v_i \sim$ i.i.d $N\left(0, \sigma_v^2\right)$.
2. $u_i \sim$ i.i.d $N^+\left(0, \sigma_u^2\right)$ (half-normal).
3. $u_i$ and $v_i$ are independents.
4. $u_i$ and $v_i$ are uncorrelated with the explanatory variables.

Since $u_i$ and $v_i$ are independents, the joint distribution of these variables is the product of their respective marginal densities:

$$f(u,v) = \frac{1}{\pi\sigma_u\sigma_v}e^{\left(-\frac{u^2}{2\sigma_u^2} - \frac{v^2}{2\sigma_v^2}\right)} \tag{6}$$

Given that $\varepsilon_i = v_i + u_i$, we obtain the joint distribution of $u_i$ and $\varepsilon_i$:

$$f(u,\varepsilon) = \frac{1}{\pi\sigma_u\sigma_v}e^{\left(-\frac{u^2}{2\sigma_u^2} - \frac{(\varepsilon-u)^2}{2\sigma_v^2}\right)} \tag{7}$$

Next, the marginal distribution of $\varepsilon_i$ is achieved by the integration of the joint density function of Equation (7):

$$f(\varepsilon) = \int_0^{Y=} f(u,\varepsilon)du = \frac{2}{\sqrt{2\pi}\sigma}\left[1 - \Phi\left(\frac{-\varepsilon\lambda}{\sigma}\right)\right] \times e^{-\frac{\varepsilon^2}{2\sigma^2}} = \frac{2}{\sigma}\varphi\left(\frac{\varepsilon}{\sigma}\right)\Phi\left(\frac{\varepsilon\lambda}{\sigma}\right) \tag{8}$$

where $\lambda = \sigma_u/\sigma_v$, $\sigma = \sqrt{\sigma_u^2 + \sigma_v^2}$ and $\phi$ and $\Phi$ are, respectively, the density and the cumulative distribution of an N(0,1).

Then, based on the probability density function (PDF) in Equation (8), we can compute the logarithm of the likelihood function for a sample with n DMUs:

$$\log(L) = \text{constant} - n\log(\sigma) + \sum_{i=1}^n \log\left(\Phi\left(\frac{\lambda\varepsilon_i}{\sigma}\right)\right) - \frac{1}{2\sigma^2}\sum_{i=1}^n \varepsilon_i^2 \tag{9}$$

The maximum likelihood estimates correspond to the values of $\sigma_u$, $\sigma_v$, and $\beta$ that maximize the Equation (9). These estimates are asymptotically consistent [17,22]. It should be noted that the logarithm of the likelihood was parameterized in terms of $\sigma^2 = \sigma_u^2 + \sigma_v^2$ and $\lambda = \sigma_u/\sigma_v$. This parameterization allowed for a better interpretation of the model, e.g., the statistic $\gamma = \sigma_u^2/\left(\sigma_u^2 + \sigma_v^2\right) = \lambda^2/\left(1 + \lambda^2\right) \in$ [0,1] allowed us to evaluate which of the two components of the compound error was predominant. In the model, $\gamma = 0$, the inefficiency is non-existent, since $\sigma_v^2$ represents the greater part of the error. In this way, the deviations between the frontier and the DMUs are random noises. Conversely, when $\gamma = 1$, the error is dominated by $\sigma_u^2$, so the deviations from the frontier are due to inefficiency. Thus, through the maximum likelihood ratio test, the hypothesis $H_0$ $\gamma = 0$ can be tested against the alternative hypothesis $H_1$ $\gamma \neq 0$ to evaluate whether the inefficiency is present in the analyzed data set. In the case of the half-normal SFA model, the same adopted in this work, the distribution of the test statistic can be approximated by a $\chi_1^2$, i.e., a chi-square distribution with 1 degree of freedom [17].

As indicated in Equation (5), in order to estimate the efficiency of each DMU, it is necessary to have an estimate of $u_i$, the error component that captures the effect of inefficiency. This estimate can be obtained from the residues because $\varepsilon_i = v_i + u_i$. By means of the joint density in Equation (7) and the density of $\varepsilon_i$ in Equation (8), we can define the conditional probability density of $u_i$ given $\varepsilon_i$:

$$f(u|\varepsilon) = \frac{f(u,\varepsilon)}{f(\varepsilon)} = \frac{1}{\sqrt{2\pi}\,\sigma_*} e^{\left[\frac{-(u-\mu_*)^2}{2\sigma_*^2}\right]} / \left[1 - \Phi\left(\frac{-\mu_*}{\sigma_*}\right)\right] u_i|\varepsilon_i \sim N^+\left(\mu_*, \sigma_*^2\right) \tag{10}$$

where $\mu_* = \frac{\varepsilon\,\sigma_u^2}{\sigma^2}$ and $\sigma_*^2 = \frac{\sigma_u^2\sigma_v^2}{\sigma^2}$.

A point estimate $\hat{u}_i$ for $u_i$ may be the mean or the mode of the conditional distribution in Equation (10). In either case, the efficiency estimate is equal to $\exp(-\hat{u}_i)$; for example, the expected value of conditional density probability $f(u_i|\varepsilon_i)$ [17,22,27] is a point estimate of $u_i$:

$$\hat{u}_i = E(u_i|\varepsilon_i) = \mu_{*i} + \sigma_*\left[\frac{\phi(-\mu_{*i}/\sigma_*)}{1 - \Phi(-\mu_{*i}/\sigma_*)}\right] = \sigma_*\left[\frac{\phi(\lambda\varepsilon_i/\sigma)}{1 - \Phi(-\lambda\varepsilon_i/\sigma)} + \frac{\lambda\varepsilon_i}{\sigma}\right] \tag{11}$$

An alternative is the efficiency measure proposed by Battese and Coelli [17]:

$$\theta_i = E(e^{-u_i}|\varepsilon_i) = \left[\frac{1 - \Phi\left(\sigma_* - \frac{\mu_{*i}}{\sigma_*}\right)}{1 - \Phi\left(-\frac{\mu_{*i}}{\sigma_*}\right)}\right] e^{-\mu_{*i} + \frac{1}{2}\sigma_*^2} \tag{12}$$

The estimation from Equation (12) differed from the estimates calculated in Equation (11). The presented results are based on the assumption that the random component $u_i$ has a half-normal distribution, but other assumptions for the probability density of $u_i$ could be admitted, e.g., truncated normal, exponential, and gamma [17,28].

### 2.2. Cost Stochastic Frontier Model for Panel Data

A sophistication introduced by the ANEEL was the specification of a panel data econometric model to estimate the socioeconomic complexity construct [16], i.e., the collection of observations of a set of variables over a period T for each one of the n distribution utilities.

Cross-section data are collected by observing many subjects, such as firms and countries, at one period of time, e.g., the set of annual balance sheets for the last year from each Brazilian distribution utilities. On the other hand, panel data contain observations of multiple variables obtained over multiple periods for the same subjects, e.g., the set of annual balance sheets for each Brazilian distribution utility over the last decade.

By incorporating the longitudinal character of the data, which makes it possible to treat any underlying correlation structures in a more appropriate way, studies based on panel data allow for the more efficient monitoring of individual units (in this case, utilities) than those based on cross section data.

Panel data have more observations than cross-section data, so they are expected to obtain more efficient estimators for the model parameters and efficiencies [17,29]. The specification of a panel data model is like the specification for the cross-sectional data model in Equation (3), but in the panel data model, the variables are indexed in time according to Equation (13):

$$\log(Y_{it}) = \beta_0 + \beta_1 \log(x_{1t}) + \ldots + \beta_k \log(x_{kt}) + v_{it} + u_{it}, \forall\, i = 1, n, \forall\, t = 1, T \tag{13}$$

Let $u_{it}$ and $v_{it}$ be independent random variables, both uncorrelated with the explanatory variables $x_{it}$; the parameters of the panel data model can be estimated in the same way as the parameters for the cross-section data model. However, the premise that $u_{it}$ are independent is unrealistic, since efficiency must vary over time as a function of technological evolution and management improvement [17,24].

Thus, it is convenient to admit some structure for the temporal evolution of the term $u_{it}$ in Equation (13), e.g., $u_{it} = f(t) \cdot u_i$, where $f(t)$ is a function that determines how efficiency evolves over time [17,29,30]. If the inefficiency effect is constant across time, then $u_{it} = u_i$ and $f(t) = 1$.

Battese and Coelli [30] proposed $f(t) = \exp[\eta(t - T)]$, where $\eta$ is a parameter to be estimated. In the panel data models, the likelihood ratio test and the z test can be applied to evaluate the hypotheses of time invariant efficiency effects $H_0$: $\eta = 0$, i.e., the inefficiency effect is constant across time [17,24]. This specification for $f(t)$ implies that the rank ordering of efficiency scores for firms remains unchanged over time [17]. In addition, in the panel data model estimation, the terms $u_{it}$ can be treated as fixed effects or random effects, with the latter option being recommended by Kumbhakar [29] and Battese and Coelli [30].

*2.3. Calculating the PNTL Targets*

For non-technical loss control purposes, a PNTL corresponds to the ratio of NTLs by low voltage market (LVM), both given in MWh:

$$\text{PNTL} = \frac{\text{NTL}}{\text{LVM}} \times 100\% \tag{14}$$

Non-technical losses can be interpreted as a cost. Thus, for a given socioeconomic complexity, the management of the utility should reduce the PNTL at the lowest possible level, i.e., the target defined by the efficiency frontier.

The idea described above is illustrated in Figure 6, in which the gray area is the possibility set for the PNTL values. Note that the set is lower bounded by the efficiency frontier defined as a function of the socioeconomic complexity in the utility's area. Thus, a utility with PNTL (Y), above the frontier, should reduce its PNTL to the target level ($Y^* < Y$) determined by the efficiency frontier. The distance to frontier reflects failures in the management of the PNTL. As such, the relative efficiency of an utility can be evaluated by the radial metric $\theta = Y^*/Y$, a number in the interval [0,1] whose complement $1-\theta$ quantifies the reduction of the PNTL to reach the target value.

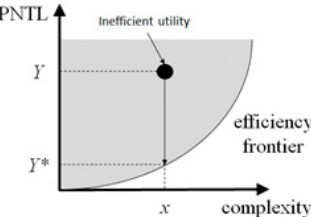

**Figure 6.** Efficiency frontier for the PNTL.

The purpose of the cost SFA model is to set the target value for the percentage of non-technical losses. The proposed model follows the Cobb–Douglas form [17]—the dependent variable is the PNTL, and the list of explanatory variables includes the same variables adopted by the ANEEL's econometric model [10,16], i.e., the proportion of subnormal households, garbage collection coverage, income inequality (Gini index), credit default level, index of violent deaths, the proportion of low-income clients in the low-voltage residential market, and the percentage of people with incomes less than half a minimum wage. Additionally, the random components $u_i$ and $v_i$ follow the half-normal and normal distributions, respectively. Here, the same specification for the random variables u and v was adopted in [23,26]. In addition, the efficiencies vary in time according to the proposal of Battese and Coelli [30].

The linearization of the Cobb–Douglas specification undergoes a logarithmic transformation; however, the PNTL is often a number in the interval [0,1] (in rare exceptions, the PNTL is equal to or greater than 1, which indicates a serious management problem), and, eventually, the PNTL can be null or very small. Then, the probability distribution of the logarithm of the dependent variable may not be compatible with the assumption of positive asymmetry for the compound error assumed by the stochastic cost frontier, as shown in Figure 7. In order to overcome this problem, we replaced the log(PNTL) with log(1 + PNTL).

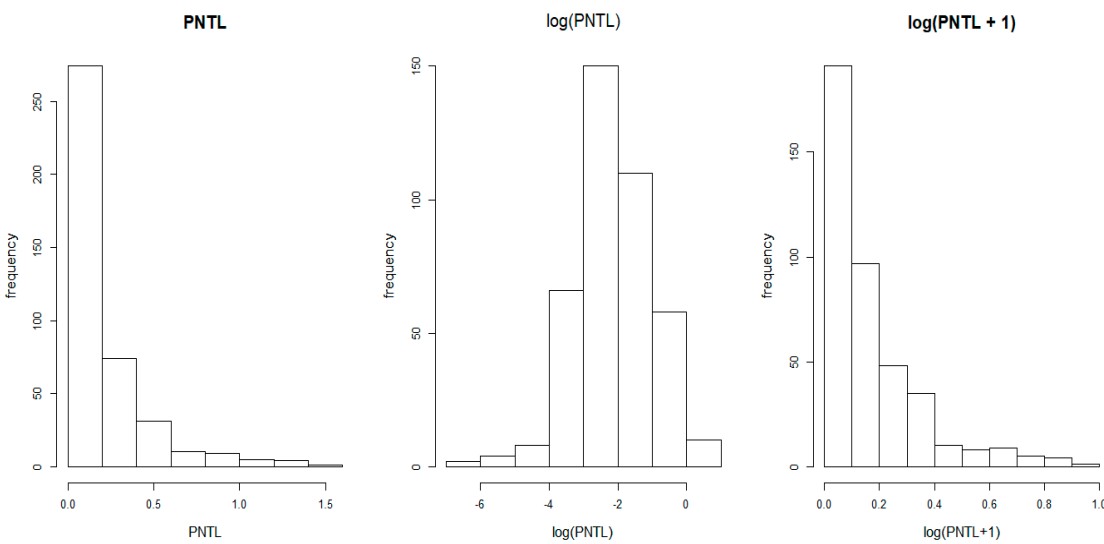

**Figure 7.** Frequency distributions of the PNTL.

The efficiency index resulting from the SFA model is denoted by $\theta$ ($\theta < 1$), so the utility's management should reduce from its current value of percentage of non-technical losses ($PNTL_0$) to the target value, i.e., the product $\theta \times PNTL_0$ at the end of a transition period $\Delta$ established by the ANEEL, e.g., the duration of a four-year tariff review cycle ($\Delta = 4$). In addition, annual intermediate target values can be established for each year t ($1 \leq t \leq \Delta$) based on the geometric rate:

$$Target_t = PNTL_0\left(\theta^{t/\Delta}\right) \tag{15}$$

## 3. Application of the Proposed Methodology

In order to illustrate the potential of the proposed methodology, the SFA model was applied on panel data with yearly observations from 2007 to 2016 (T = 10) and for 41 (n = 41) Brazilian electricity distribution utilities; therefore, the analyzed period covered two tariff review cycles. It is noteworthy that there were missing observations in two evaluated utilities, one missing observation in each one. Below, Figure 8 shows the spatial distribution of the PNTLs among the Brazilian states at 2016. It is worth mentioning the existence of many isolated systems in the states located in the Amazon region. Next, Figure 9 shows the trajectories of the PNTLs of the evaluated utilities during the period of 2007–2016.

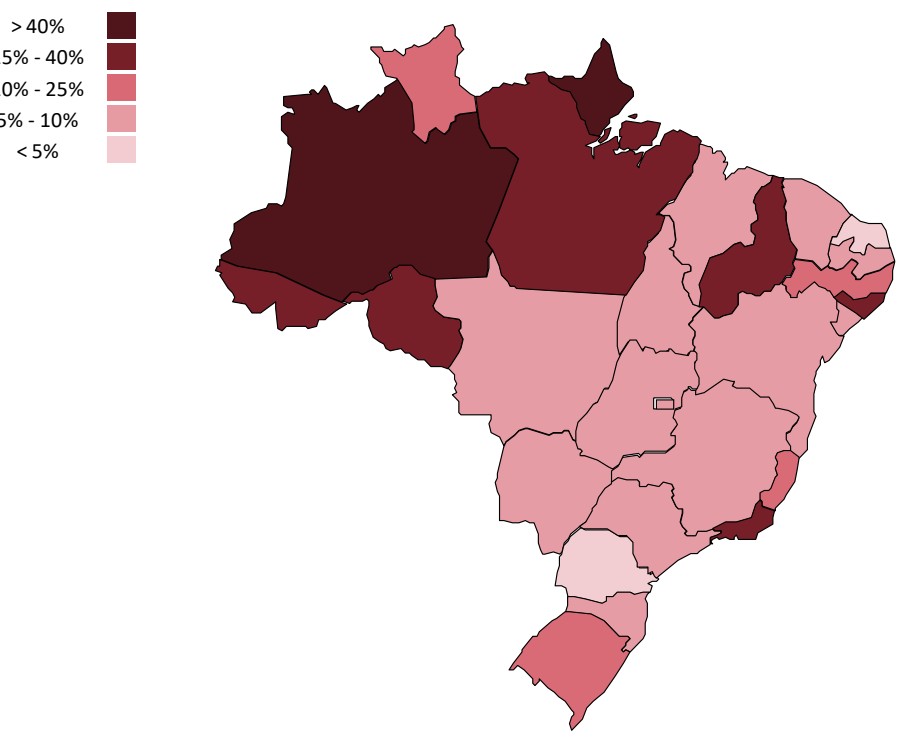

**Figure 8.** Spatial distribution of the PNTLs among the Brazilian states in 2016 [9].

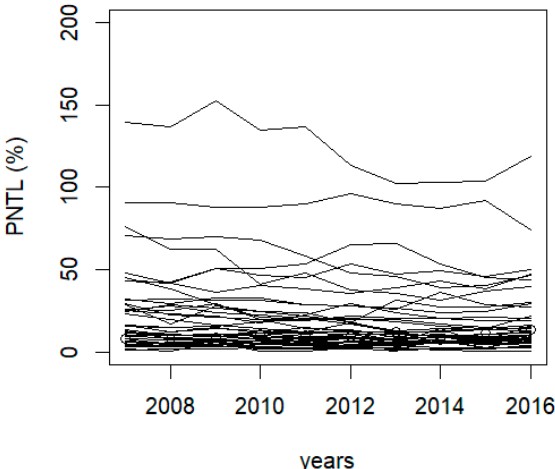

**Figure 9.** Trajectories of the PNTLs in low voltage (LV) networks [9].

Figure 9 shows the trajectories of the PNTLs in LV networks [9], where the variability reflects the huge heterogeneity in the socioeconomic complexity and the different strategies for the non-technical losses management in the Brazilian distribution utilities; for example, the biggest losses were observed in a few companies that serve isolated systems that are not connected to the distribution network.

The explanatory variables used in the application of SFA are the same seven considered in the panel data econometric model specified by the ANEEL. One associated with violence (vio), four associated with poverty/income inequality (Gini, default, lowinc, and poor), and two associated with the infrastructure quality (sub and garbage). A brief explanation of the variables can be found in Table 1.

**Table 1.** The explanatory variables used in the application of stochastic frontier analysis (SFA) and the panel data econometric model specified by the National Electric Energy Agency (ANEEL) [10].

| Variable | Definition | Unit/Range | Source |
|---|---|---|---|
| Vio | Number of violent deaths | Deaths per 100 thousand inhabitants | Brazil's National Health System |
| Garbage | People served by garbage collection services | % over the population | Brazilian National Bureau of statistic |
| Poor | People with per capita income below 1/2 minimum wage | % over the population | |
| Gini | Degree of income inequality | 0–1 (1 high inequality—0 low inequality) | |
| Sub | Subnormal households | % over the total households | |
| Default | Credit sector default | % default over total credit | Brazilian Central Bank |
| Lowinc | Household energy consumption covered by the Electricity Social Tariff (low income subsidy) | % over Total Household energy consumption | ANEEL |

The computational implementation was performed in the R programming language [24,31] and based on the plm [32] and frontier [17] packages. The plm package organizes data in a panel data framework, and the frontier package estimates the SFA model by maximum likelihood method. The regression coefficients, gamma ($\gamma$), and time ($\eta$) statistics are presented in Figure 10.

```
Error Components Frontier (see Battese & Coelli 1992)
Inefficiency increases the endogenous variable (as in a cost function)
The dependent variable is logged
Iterative ML estimation terminated after 29 iterations:
log likelihood values and parameters of two successive iterations
are within the tolerance limit
final maximum likelihood estimates
             Estimate Std. Error  z value  Pr(>|z|)
(Intercept)  0.1755888  0.0681187    2.5777  0.009946 **
log(vio)     0.0030785  0.0090805    0.3390  0.734591
log(lixo)   -0.0844971  0.0302439   -2.7939  0.005208 **
log(pob)     0.0033015  0.0146176    0.2259  0.821312
log(gini)    0.0846581  0.0466296    1.8155  0.069440 .
log(sub)     0.0211949  0.0122162    1.7350  0.082745 .
log(inad)    0.0209452  0.0113499    1.8454  0.064977 .
log(mb1mbr) -0.0186176  0.0064014   -2.9084  0.003633 **
sigmaSq      0.0327465  0.0079323    4.1283 3.655e-05 ***
gamma        0.9647026  0.0090837  106.2014 < 2.2e-16 ***
time         0.0239479  0.0042054    5.6946 1.237e-08 ***
Signif. codes:  0 '***' 0.001 '**' 0.01 '*' 0.05 '.' 0.1 ' ' 1
log likelihood value: 708.2648
panel data
number of cross-sections = 41
number of time periods = 10
total number of observations = 408
thus there are 2 observations not in the panel
mean efficiency of each year
     2007      2008      2009      2010      2011      2012      2013      2014
0.8702755 0.8729316 0.8755416 0.8732528 0.8806248 0.8830991 0.8855291 0.8879153
     2015      2016
0.8902584 0.8925587
mean efficiency: 0.8812376
```

**Figure 10.** Fitted model.

As shown in Figure 10, the estimate of the gamma parameter ($\gamma$) was statistically different from zero and assumed value of 0.96, very close to 1; therefore, inefficiency was present. In addition, the estimation of the time effect ($\eta$) was also statistically significant and assumed a positive value,

i.e., efficiencies increased over time, as indicated by the mean efficiency of each year in Figure 10. The estimated efficiencies for each one of the 41 utilities are presented in Appendix A and Figure 11, where the line in black is the average and the variability reflects the huge heterogeneity in the current levels of the non-technical losses of the Brazilian distribution utilities.

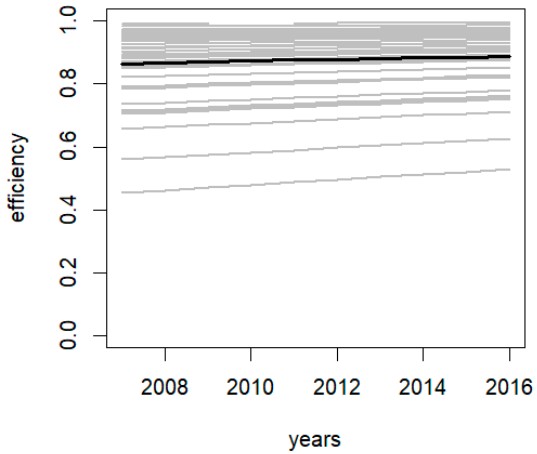

**Figure 11.** Estimated efficiencies for the period of 2007–2016.

## 4. Results and Discussion

The results shown in Figure 10 also indicate that the coefficients (elasticities) of the "violent death rate" (vio) and the "percentage of people with incomes less than half a minimum wage" (poor) were not statistically significant at the usual levels of significance of 1%, 5%, and 10%. The coefficients of the "collection of garbage" (garbage) and the "proportion of low-income customers in the low-voltage residential market" (lowinc) appeared with negative coefficients. Thus, the increase in these variables reduced the PNTL. The negative coefficient for the "lowinc" variable suggested that the discounts on electricity bill for the low-income families contributed to reducing the PNTL. In a similar way, the negative coefficient for the "garbage" variable indicated that better delivery conditions (lower socioeconomic complexity) could reduce the PNTL. On the other hand, the results showed that the income inequality, subnormal households, and credit default (socioeconomic complexity) contributed to increasing the PNTL.

From the efficiencies estimates, we could take the median efficiency for each utility in order to define the respective PNTL target value. Table 1 presents the targets for the PNTL determined by the SFA model, taking the first decile of the PNTL in the period of 2007–2016 as the base value. Figure 12 shows the PNTL boxplots over the period of 2007–2016 for each utility, with the respective target values indicated by a solid black line. Note that in some utilities, the target values fell within the range defined by the respective boxplots or were slightly below the lower fences. Therefore, the targets determined by the SFA model were feasible and could be achieved by the utilities. Additionally, in the utilities with high PNTL levels, the targets were more aggressive and were far from boxplots. More aggressive targets could be achieved by reducing the base value in Table 2.

In order to allow for comparisons with targets from the ANEEL's model and the verified PNTL in 2018, the targets from the SFA model (Table 2) were updated to 2018 via Equation (15). The targets from SFA for 2018 are presented in Table 3. Next, Figure 13 shows the targets from the ANEEL's and SFA models for 2018.

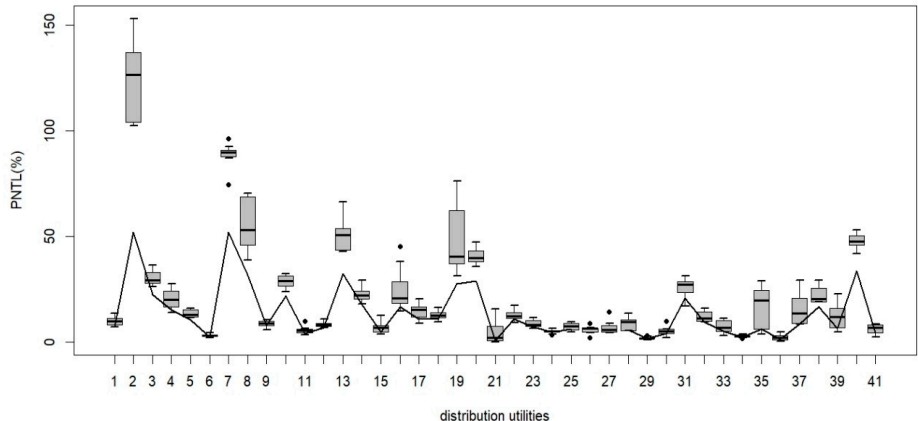

**Figure 12.** Boxplots of the PNTLs over the period of 2007–2016 and target values.

**Table 2.** Targets.

| Utility | PNTL Base Value (A) | Median Efficiency θ (B) | PNTL Target Value (C = A × B) | Utility | PNTL Base Value (A) | Median Efficiency θ (B) | PNTL Target Value (C = A × B) |
|---|---|---|---|---|---|---|---|
| 1 | 7.7% | 92.2% | 7.1% | 22 | 10.4% | 95.1% | 9.9% |
| 2 | 103.1% | 49.1% | 50.6% | 23 | 7.0% | 97.0% | 6.8% |
| 3 | 27.3% | 80.5% | 22.0% | 24 | 3.7% | 97.2% | 3.6% |
| 4 | 14.5% | 87.7% | 12.7% | 25 | 5.2% | 96.8% | 5.0% |
| 5 | 12.0% | 88.3% | 10.6% | 26 | 4.0% | 99.3% | 4.0% |
| 6 | 2.3% | 94.2% | 2.2% | 27 | 4.3% | 95.5% | 4.1% |
| 7 | 85.8% | 59.3% | 50.9% | 28 | 5.5% | 95.9% | 5.3% |
| 8 | 40.0% | 68.5% | 27.4% | 29 | 1.2% | 92.5% | 1.1% |
| 9 | 6.2% | 97.6% | 6.1% | 30 | 3.2% | 97.1% | 3.1% |
| 10 | 24.6% | 81.0% | 19.9% | 31 | 17.3% | 86.4% | 14.9% |
| 11 | 3.7% | 96.1% | 3.6% | 32 | 9.9% | 94.3% | 9.3% |
| 12 | 7.1% | 89.7% | 6.4% | 33 | 3.9% | 96.4% | 3.7% |
| 13 | 43.3% | 73.5% | 31.8% | 34 | 1.9% | 96.0% | 1.8% |
| 14 | 20.1% | 86.9% | 17.5% | 35 | 4.8% | 83.8% | 4.0% |
| 15 | 3.8% | 95.1% | 3.6% | 36 | 0.3% | 94.7% | 0.3% |
| 16 | 15.2% | 89.3% | 13.6% | 37 | 8.5% | 90.0% | 7.6% |
| 17 | 10.8% | 90.4% | 9.8% | 38 | 19.0% | 86.8% | 16.5% |
| 18 | 10.6% | 91.3% | 9.7% | 39 | 6.4% | 93.4% | 6.0% |
| 19 | 35.1% | 74.0% | 26.0% | 40 | 44.9% | 72.9% | 32.8% |
| 20 | 36.2% | 75.8% | 27.5% | 41 | 3.5% | 95.7% | 3.4% |
| 21 | 0.3% | 98.5% | 0.3% | Median | 7.7% | 92.2% | 7.1% |

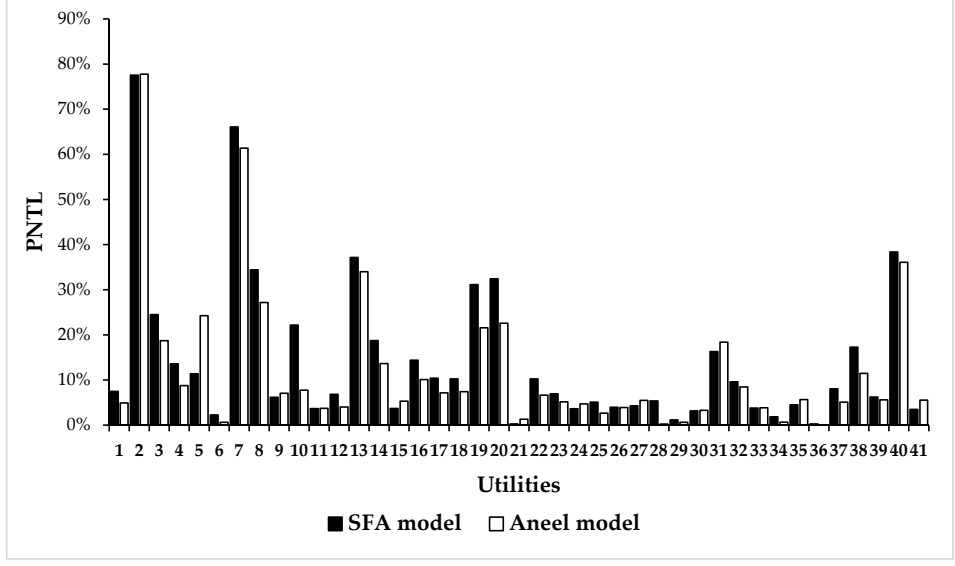

**Figure 13.** PNTL targets from the SFA model and the ANEEL's model for 2018 [9].

Table 3. Targets from SFA for 2018.

| Utility | Year Tariff Review Cycle ($\Delta$) | PNTL Base Value (A) | Median Efficiency $\theta$ (B) | PNTL Target Value 2018 (C $= A \times B^{2/\Delta}$) | Utility | Year Tariff Review Cycle ($\Delta$) | PNTL Base Value (A) | Median Efficiency $\theta$ (B) | PNTL Target Value 2018 (C $= A \times B^{2/\Delta}$) |
|---|---|---|---|---|---|---|---|---|---|
| 1 | 5 | 7.7% | 92.2% | 7.46% | 22 | 4 | 10.4% | 95.1% | 10.23% |
| 2 | 5 | 103.1% | 49.1% | 77.56% | 23 | 5 | 7.0% | 97.0% | 6.93% |
| 3 | 4 | 27.3% | 80.5% | 24.48% | 24 | 4 | 3.7% | 97.2% | 3.62% |
| 4 | 4 | 14.5% | 87.7% | 13.58% | 25 | 4 | 5.2% | 96.8% | 5.10% |
| 5 | 5 | 12.0% | 88.3% | 11.38% | 26 | 5 | 4.0% | 99.3% | 3.97% |
| 6 | 4 | 2.3% | 94.2% | 2.26% | 27 | 4 | 4.3% | 95.5% | 4.27% |
| 7 | 4 | 85.8% | 59.3% | 66.08% | 28 | 5 | 5.5% | 95.9% | 5.36% |
| 8 | 5 | 40.0% | 68.5% | 34.41% | 29 | 4 | 1.2% | 92.5% | 1.17% |
| 9 | 4 | 6.2% | 97.6% | 6.15% | 30 | 4 | 3.2% | 97.1% | 3.15% |
| 10 | 4 | 24.6% | 81.0% | 22.16% | 31 | 4 | 17.3% | 86.4% | 16.28% |
| 11 | 4 | 3.7% | 96.1% | 3.67% | 32 | 5 | 9.9% | 94.3% | 9.59% |
| 12 | 5 | 7.1% | 89.7% | 6.81% | 33 | 4 | 3.9% | 96.4% | 3.79% |
| 13 | 4 | 43.3% | 73.5% | 37.14% | 34 | 4 | 1.9% | 96.0% | 1.83% |
| 14 | 4 | 20.1% | 86.9% | 18.74% | 35 | 4 | 4.8% | 83.8% | 4.49% |
| 15 | 4 | 3.8% | 95.1% | 3.68% | 36 | 5 | 0.3% | 94.7% | 0.27% |
| 16 | 4 | 15.2% | 89.3% | 14.38% | 37 | 4 | 8.5% | 90.0% | 8.04% |
| 17 | 5 | 10.8% | 90.4% | 10.38% | 38 | 4 | 19.0% | 86.8% | 17.28% |
| 18 | 5 | 10.6% | 91.3% | 10.25% | 39 | 3 | 6.4% | 93.4% | 6.21% |
| 19 | 5 | 35.1% | 74.0% | 31.14% | 40 | 5 | 44.9% | 72.9% | 38.36% |
| 20 | 5 | 36.2% | 75.8% | 32.41% | 41 | 4 | 3.5% | 95.7% | 3.48% |
| 21 | 4 | 0.3% | 98.5% | 0.25% | | Median | 7.7% | 92.2% | 10.23% |

In Figure 13, the targets calculated by the ANEEL's model to 2018 can be seen to have been more stringent than the targets from SFA approach in 28 of the 41 analyzed utilities. In addition, the verified PNTLs in 2018 were above that of the targets from the ANEEL's model in 34 utilities, while only 22 utilities presented PNTLs above the targets from the SFA model for the same year. As illustrated in Table 4, only seven utilities achieved the targets defined by the ANEEL's model, while 19 utilities achieved the targets from the SFA approach. In 20 utilities, the verified PNTLs were greater than the targets defined by both methodologies.

**Table 4.** Deviations between the targets from the ANEEL's model and targets from SFA for 2018.

| Absolute Deviation between Targets from SFA and ANEEL | Frequency | Utilities |
|---|---|---|
| 0–1% | 9 | 2 [a], 9 [a], 11 [a], 26 [a], 29 [a], 30 [a], 33 [a], 36 [b,c], 39 [b,c] |
| 1–2% | 9 | 6 [b], 15 [c], 21 [b,c], 23 [a], 24 [a], 27 [a], 32 [b], 34 [a], 35 [a] |
| 2–3% | 8 | 1 [a], 12 [b], 18 [b], 25 [b,c], 31 [c], 37 [b], 40 [a], 41 [a] |
| 3–4% | 3 | 13 [a], 17 [b], 22 [b] |
| 4–5% | 3 | 4 [b], 7 [a], 16 [b,c] |
| >5% | 9 | 3 [a], 5 [a], 8 [b], 10 [a], 14 [b], 19 [b], 20 [b], 28 [b], 38 [b] |

Note: [a] is for utilities that not meet both targets, [b] is for utilities that met their SFA targets, and [c] is for utilities that met the targets set by the ANEEL.

The largest difference was of the order of 14% at utility 10, for which the ANEEL's model suggested a reduction of the PNTL to almost 8%, while the SFA model suggested a reduction of approximately 22%. Since the base value adopted in the work was the first decile of the PNTL in the period of 2007–2016 (24.6% for utility 10), we observed that the target defined by the ANEEL for utility 10 was unattainable. In fact, in 2018, the verified PNTL in utility 10 (22.9%) was very close to the target of 22% set by the SFA model, but it was way above the target of 8% from the ANEEL's model. The same situation was observed in utilities 19 and 20, as illustrated in Figure 13. It is worth pointing out that unattainable targets compromised the economic balance of the utilities. The ANEEL's approach assigned targets of less than 1% for the PNTLs of utilities 6, 28, 29, 34, and 36 (five utilities), while in the SFA approach, only utilities 21 and 36 had targets below 1%, and both presented PNTL below the targets.

On the other hand, Figure 13 shows a different situation in utility 5, where the target from the ANEEL's model (24.2%) was almost the double of the target from the SFA approach (11.4%). It is important to highlight that the target value defined by the ANEEL for utility 5 in 2018 was way above the PNTL values in the period of 2007–2016. One possible explanation for the high target value (24.2%) was the deterioration in the supply conditions of utility 5, which caused losses in 2018 (almost 29%) above historical levels in 2007–2016. In addition, utility 5 went through a privatization process. Therefore, the high target for company 5 may have reflected decisions from the discretionary power of the regulatory agency.

The largest deviations (>5%) between the results from models were in utilities 3, 5, 8, 10, 14, 19, 20, 28, and 38, as indicated by the bottom row of the frequency distribution in Table 4. In this group, the targets defined by the ANEEL model tended to be lower than the targets from the SFA approach, and they were not achieved in 2018, as indicated in Table 4. In some of these cases, the methodology employed by the ANEEL required a reduction in the NTL that was not justifiable for distributors that already had very low NTL levels. For example, utility 28 registered a very low NTL level in 2018 0.54%); however, it was above the target set by the ANEEL (0.24%). The target set by the SFA was 5.36% and therefore avoided an improper penalty of the utility.

In others cases, the methodology used by the ANEEL did not seem to correctly consider the historical performance of utilities in reducing NTLs. For example, utilities 3, 8, 14, and 19 operate in extremely complex areas (high levels of violence and social inequality); though they continue to register a high level of NTLs, these utilities have had a very positive history of NTL reductions over

the past few years. In 2018, all these utilities had NTLs higher than the regulatory NTL target set by the ANEEL, which implied a financial loss for these companies. However, if these companies were compared with the goals established from the SFA, they would all have had NTL levels lower than or very close to the target. In recent years, the agency itself has recognized the limitation of his own model in relation to these utilities and has, in different ways, adjusted the regulatory target of these utilities.

Despite the differences, in nine utilities, the deviations were lower than 1%, and the two models proposed similar targets for utility 2, where the largest PNTL was observed in 2018 (almost 115%, when considering billed energy consumption). Additionally, the Pearson and Spearman correlation coefficients were 0.97 and 0.95, respectively. In addition, the targets defined by the ANEEL's model for utilities 8, 10, 19, 20, and 38 could probably not be reached.

It is important to note that the objective of this session was only to show the feasibility of applying the proposed methodology and its possible benefits—hence the selection of only one year (2018). A deeper analysis of the NTL behavior of Brazilian distributors should cover a broader period.

## 5. Conclusions

Electric regulatory agencies worldwide have large experience with benchmarking methods—DEA and SFA in particular, which have been applied in the regulation of transmission and distribution sectors. For example, the ANEEL has adopted DEA models to assessment the efficiency levels of the operation expenditures for transmission and distribution utilities.

The theft of electric power is a problem faced by Brazilian distribution utilities. Aiming to guide the distribution utilities in combating NTLs, the ANEEL has adopted a regulatory strategy based on the principles of benchmarking. The ANEEL's strategy is implemented by a panel data econometric model that provides target values for the NTLs for each utility. The econometric approach adopted by the ANEEL follows some basic principles of benchmarking and yardstick competition present in the DEA and SFA models, but it fails because it does not have a clear definition of the efficiency frontier, the main component in a benchmarking framework. In order to overcome this deficiency, we formulated the ANEEL's econometric model like an SFA model in this work.

We highlighted that the option for SFA models maintained the econometric framework initially adopted by the ANEEL, i.e., the same dependent and explanatory variables in a panel data model. However, the SFA formulation allowed us to estimate the efficiency frontier, a tool that provides PNTL target values in a more transparent way (the current methodology adopted by the ANEEL employs complex criteria that require further clarification). The use of SFA makes the regulatory procedure transparent and reproducible.

The SFA model can take different forms. We evaluated other specifications, but the most satisfactory results were produced by the cost SFA model with the Cobb–Douglas equation and inefficiency term error with a half-normal distribution, i.e., the basic normal/half-normal SFA model.

Finally, the results from the case study with the main Brazilian electric distribution utilities showed that the proposed cost SFA model could provide feasible target values for the PNTL, i.e., targets that could be reached by the distribution utilities and that satisfy a range of economic, social, and political constraints while also keeping the focus on controlling non-technical losses.

**Author Contributions:** Conceptualization, J.P., R.S., R.C. and P.S.; Methodology, J.P., D.L. and P.S., software, J.P. and P.S.; validation, D.L. and R.C.; formal analysis, J.P., D.L. and R.C.; investigation, J.P., P.S. and D.L.; resources, R.S., R.C. and D.L.; data curation, D.L. and R.C.; writing—original draft preparation, D.L., J.P. and R.C.; writing—review and editing, D.L., J.P., P.S., R.C. and R.S.; supervision, R.C. and R.S.; project administration, R.C. and R.S.; funding acquisition, R.C. and R.S. All authors have read and agreed to the published version of the manuscript.

**Funding:** Financial support of R&D program of the Brazilian Electricity Regulatory Agency (ANEEL)-R&D project PD 00383-0062/2017. This study was financed in part by the Coordenação de Aperfeiçoamento de Pessoal de Nível Superior-Brasil (CAPES)-Finance Code 001.

**Acknowledgments:** The authors would like to thank the colleagues from PUC-Rio for their valuable comments and suggestions, which improved the paper, and the R&D program of the Brazilian Electricity Regulatory Agency (ANEEL) for the financial support (R&D project PD 00383-0062/2017).

**Conflicts of Interest:** The authors declare no conflict of interest.

## Appendix A

**Table A1.** Efficiencies calculated with the SFA model.

| Utility | 2007 | 2008 | 2009 | 2010 | 2011 | 2012 | 2013 | 2014 | 2015 | 2016 |
|---|---|---|---|---|---|---|---|---|---|---|
| 1 | 91.4% | 91.6% | 91.8% | 91.9% | 92.1% | 92.3% | 92.5% | 92.6% | 92.8% | 93.0% |
| 2 | 45.3% | 46.1% | 47.0% | 47.8% | 48.7% | 49.5% | 50.4% | 51.2% | 52.0% | 52.8% |
| 3 | 78.5% | 78.9% | 79.4% | 79.8% | 80.2% | 80.7% | 81.1% | 81.5% | 81.9% | 82.3% |
| 4 | 86.4% | 86.7% | 87.0% | 87.2% | 87.5% | 87.8% | 88.1% | 88.3% | 88.6% | 88.9% |
| 5 | 87.1% | 87.4% | 87.6% | 87.9% | 88.2% | 88.5% | 88.7% | 89.0% | 89.2% | 89.4% |
| 6 | 93.6% | 93.7% | 93.9% | 94.0% | 94.1% | 94.3% | 94.4% | 94.5% | 94.7% | 94.8% |
| 7 | 55.9% | 56.7% | 57.4% | 58.2% | 59.0% | 59.7% | 60.4% | 61.2% | 61.9% | 62.6% |
| 8 | 65.6% | 66.3% | 66.9% | 67.6% | 68.2% | 68.8% | 69.4% | 70.0% | 70.6% | 71.2% |
| 9 | 97.3% | 97.3% | 97.4% | 97.5% | 97.5% | 97.6% | 97.6% | 97.7% | 97.8% | 97.8% |
| 10 | 79.1% | 79.6% | 80.0% | 80.4% | 80.8% | 81.3% | 81.7% | 82.0% | 82.4% | 82.8% |
| 11 | 95.7% | 95.8% | 95.9% | 96.0% | 96.0% | 96.1% | 96.2% | 96.3% | 96.4% | 96.5% |
| 12 | 88.6% | 88.8% | 89.1% | 89.3% | 89.5% | 89.8% | 90.0% | 90.2% | 90.4% | 90.7% |
| 13 | 71.0% | 71.5% | 72.1% | 72.7% | 73.2% | 73.8% | 74.3% | 74.8% | 75.3% | 75.8% |
| 14 | 85.5% | 85.9% | 86.2% | 86.5% | 86.8% | 87.1% | 87.4% | 87.6% | 87.9% | 88.2% |
| 15 | 94.5% | 94.6% | 94.8% | 94.9% | 95.0% | 95.1% | 95.2% | 95.3% | 95.4% | 95.6% |
| 16 | 88.2% | 88.5% | 88.7% | 89.0% | 89.2% | 89.4% | 89.7% | 89.9% | 90.1% | 90.4% |
| 17 | 89.3% | 89.6% | 89.8% | 90.0% | 90.3% | 90.5% | 90.7% | 90.9% | 91.1% | 91.3% |
| 18 | 90.3% | 90.6% | 90.8% | 91.0% | 91.2% | 91.4% | 91.6% | 91.8% | 92.0% | 92.1% |
| 19 | 71.5% | 72.0% | 72.6% | 73.2% | 73.7% | 74.2% | 74.8% | 75.3% | 75.8% | 76.3% |
| 20 | 73.5% | 74.0% | 74.5% | 75.1% | 75.6% | 76.1% | 76.6% | 77.1% | 77.5% | 78.0% |
| 21 | 98.3% | 98.4% | 98.4% | 98.5% | 98.5% | 98.5% | 98.6% | 98.6% | 98.6% | 98.7% |
| 22 | 94.6% | 94.7% | 94.8% | 94.9% | 95.1% | 95.2% | 95.3% | 95.4% | 95.5% | 95.6% |
| 23 | 96.7% | 96.8% | 96.8% | 96.9% | 97.0% | 97.1% | 97.1% | 97.2% | 97.3% | 97.3% |
| 24 | 96.9% | 97.0% | 97.1% | 97.1% | 97.2% | 97.3% | 97.3% | 97.4% | 97.4% | 97.5% |
| 25 | 96.4% | 96.5% | 96.6% | 96.7% | 96.7% | 96.8% | 96.9% | 97.0% | 97.0% | 97.1% |
| 26 | 99.2% | 99.2% | 99.3% | | 99.3% | 99.3% | 99.3% | 99.3% | 99.4% | 99.4% |
| 27 | 94.9% | 95.0% | 95.2% | | 95.4% | 95.5% | 95.6% | 95.7% | 95.8% | 95.9% |
| 28 | 95.4% | 95.5% | 95.6% | 95.7% | 95.8% | 95.9% | 96.0% | 96.1% | 96.2% | 96.3% |
| 29 | 91.7% | 91.9% | 92.1% | 92.2% | 92.4% | 92.6% | 92.8% | 92.9% | 93.1% | 93.2% |
| 30 | 96.7% | 96.8% | 96.9% | 97.0% | 97.0% | 97.1% | 97.2% | 97.2% | 97.3% | 97.4% |
| 31 | 85.0% | 85.3% | 85.6% | 85.9% | 86.2% | 86.5% | 86.8% | 87.1% | 87.4% | 87.7% |
| 32 | 93.6% | 93.8% | 93.9% | 94.0% | 94.2% | 94.3% | 94.5% | 94.6% | 94.7% | 94.8% |
| 33 | 96.0% | 96.1% | 96.2% | 96.3% | 96.4% | 96.5% | 96.5% | 96.6% | 96.7% | 96.8% |
| 34 | 95.6% | 95.7% | 95.8% | 95.9% | 96.0% | 96.1% | 96.2% | 96.3% | 96.3% | 96.4% |
| 35 | 82.2% | 82.5% | 82.9% | 83.3% | 83.6% | 84.0% | 84.3% | 84.7% | 85.0% | 85.3% |
| 36 | 94.1% | 94.2% | 94.3% | 94.5% | 94.6% | 94.7% | 94.9% | 95.0% | 95.1% | 95.2% |
| 37 | 89.0% | 89.2% | 89.4% | 89.7% | 89.9% | 90.1% | 90.4% | 90.6% | 90.8% | 91.0% |
| 38 | 85.4% | 85.7% | 86.0% | 86.3% | 86.6% | 86.9% | 87.2% | 87.5% | 87.8% | 88.0% |
| 39 | 92.6% | 92.8% | 93.0% | 93.1% | 93.3% | 93.4% | 93.6% | 93.7% | 93.9% | 94.0% |
| 40 | 70.4% | 71.0% | 71.5% | 72.1% | 72.7% | 73.2% | 73.8% | 74.3% | 74.8% | 75.3% |
| 41 | 95.3% | 95.4% | 95.5% | 95.6% | 95.7% | 95.8% | 95.9% | 96.0% | 96.1% | 96.2% |

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
