# Peer review of "A Stochastic Frontier Model for Definition of Non-Technical Loss Targets"

_energies, doi:10.3390/en13123227_

Round 1

Reviewer 1 Report

This is a well written and interesting study about the application of a stochastic frontier cost model for panel data, applied to the tolerable limits for the percentage of NTL of 41 distribution utilities in the Brazilian electrical system. The paper concludes that previous models used by ANEEL provided unreachable target values in many cases, but that their new model gives much more realistic values. The statistical reasoning is sound and the conclusions are supported by the data. Especially valuable is the study of the influence of the explanatory variables at the beginning of the Discussion. I have not seen any serious error or flaw which could compromise the study, the Introduction is very informative and the text can be read effortlessly. I recommend a minor style and punctuation revision as usual  for the final version (some minor comma displacements, for example).

Author Response

Response to Reviewer 1 Comments

We would like to express our thanks to the reviewer for carefully examining our manuscript and for his/her constructive comments. We have addressed all comments and suggestions and modified the manuscript accordingly.

Point 1: I recommend a minor style and punctuation revision as usual for the final version (some minor comma displacements, for example).

Response 1: A general review of the paper has been done in order to correct all these problems.

Reviewer 2 Report

The work addresses the problem of the reduction of the so called non-technical losses by suitable definition of target values by the Authority, giving a theoretical framework to model and solve such problem.

The proposed model and the discussion are sound, but the paper would benefit of some additional considerations, especially to justify the selected approach and to make teh subject clear for non experts.

1) Some English form issues:
a) line 45: "another" not "other"
b) line 68: absence of presence is viable in Latin languages, but the sentence must be reworded in English
c) Line 109. "That" not "it".
d) line 337. "Of" or "in" not "to" 2018. Line 294, as a consequence, should say "of the same year".
e) Line 385. "Particular" instead of "special".
f) Line 389. "Other" should be replaced by "another" or omitted, saying "... is a problem ..." (the latter is preferred).
g) Line 394. Present , not presents.

2) Introduction. Cross-section and panel data need to be better described and more extensively defined and characterized, with suitable references for to understand how the problem is approached.
Please clarify what are the pitfalls of linear programming and why you have selected a SFA approach. Include also an overview of successful applications of these methods to similar problems, such as energy market, optimization of energy production, etc.

3) Line 154-155. Please, clarify the suitability of the assumption that the production function is linear in the logarithms, and to which real/practical cases it can be applied.
"Cobb-Douglas" needs suitable biblio references to be well understood.

4) Sec. 2.1 & 2.2. Uniform notation for logarithm. A notation like log( ) should be preferred, having indicated if the log is natural or decimal (or other).

5) Line 201-204. Please, consider the problem of a very small value of gamma that is nearly 0 but not exactly =0: how is it considered? You should specify a threshold T, with gamma <=T and gamma>T, rather than two hard cases gamma=0 and gamma>0. T may be as little as needed to justify your definition of "0".
Moreover, how the fact that gamma is in the interval [0,1] implies that the distribution is a chi-square? Please, include suitable references. And clarifies also the notation where the chi has a subscript "1".

6) Sec. 2.2. It is important that "cross-section" and "panel" terms are well and comprehensively described now that here are compared and contrasted.

7) Line 227. Sentence is quite difficult to understand, using twice "independents" without clarifying what they are: independent variables, maybe? English form must be improved to make this sentence, and other similar ones, readable and understandable.

8) Line 235-236. Please, clarify what is the meaning of the sentence, in particular how the likelihood test is applied, what "time invariance hypothesis of the efficiency eta =0 means, and where in ref [6] this is demonstrated or clarified. Consider that ref [6] is not sufficient to demonstrate it and needs to be integrated by supplemental references.

9) Sec. 2.1 & Line 274. Ambiguous notation where we do not distinguish the product of theta and PNTL0 from an entire unique variable named "thetaPNTL0". Consider how Ln and ln (that are functions or operators) were used before, without round parentheses but stick to the variables as here at line 274 where you mean really a product, instead.

10) Fig. 9 and Fig. 11. Add in the figure caption the reason of variability (or in other words the parametrization) of the many curves.

11) Results & Discussion. The variables "lowinc","poor","garbage", etc. are only introduced in Results, and they are followed by some interpretation in Discussion: it is in this moment that the reader understands what they refer to. Instead at line 293 when referring to ANEEL the quantities should be defined with some considerations that help clarifying how they will be used, including their range of variation.

12) Line 343-377. Could you attempt some interpretations and clarification of the results, in particular why some utilities show particular behavior? Eg for utilities at lines 367 and onward, as you have done for utility 5 in lines above.

13) To the aim of a better clarification of behavior, and to account for special events (such as the losses of utility 5 occurred in 2018), you should propose a verification based on the data of a different year, such as 2017, in order to derive more general judgments.

Author Response

Response to Reviewer 2 Comments

We would like to express our thanks to the reviewer for carefully examining our manuscript and for his/her constructive comments. We have addressed all comments and suggestions and modified the manuscript accordingly.

Point 1: Some English form issues:

  1. Line 45: "another" not "other"
  2. Line 68: absence of presence is viable in Latin languages, but the sentence must be reworded in English
  3. Line 109. "That" not "it".
  4. Line 337. "Of" or "in" not "to" 2018. Line 294, as a consequence, should say "of the same year".
  5. Line 385. "Particular" instead of "special".
  6. Line 389. "Other" should be replaced by "another" or omitted, saying "... is a problem ..." (the latter is preferred).
  7. Line 394. Present , not presents.

Response 1: All suggested changes were done.

Point 2:

(a) Cross-section and panel data need to be better described and more extensively defined and characterized, with suitable references for to understand how the problem is approached.

Response 2 (a): In econometrics, cross-section and panel data are well-known ways of organizing data and for this reason, they have not been explained in the text. However, the following paragraph was included in section 2.2 of the paper (lines 253-260):

“Cross-section data is collected by observing many subjects, such as firms and countries, at the one period of time, for example, the set of annual balance sheets for the last year from each Brazilian distribution utilities. On the other hand, panel data contain observations of multiple variables obtained over multiple periods for the same subjects, for example, the set of annual balance sheets for each Brazilian distribution utility over the last decade.

By incorporating the longitudinal character of the data, making possible to treat any underlying correlation structures in a more appropriate way, studies based on panel data allow for more efficient monitoring of individual units (in this case, utilities) than those based on cross section data.”

(b) Please clarify what are the pitfalls of linear programming and why you have selected a SFA approach.

Response 2 (b): The answer is in the section 1. Introduction, the following text appears between lines 116 and 122: “The DEA and SFA approaches aim to estimate an efficiency frontier from data, but they differ in the methods employed, DEA is a non-parametric approach based on linear programming [23], while SFA is a parametric approach that relies on econometric modeling [24]. Additionally, in the DEA approach the effort undertaken by an utility to reach the benchmark (frontier) corresponds to the utility's deviation from the efficiency frontier. On the other hand, in the SFA approach there is the recognition that part of the deviation from the frontier is due to factors not manageable by the utilities [22,24], a premise compatible with the reality faced by utilities in combating non-technical losses.”

Before at line 42 appears “…NTL depends on factors not manageable by the utilities.”

In addition, at line 126 appears “The choice of the SFA approach is due to recognition that non-technical losses are determined by variables not manageable by the utilities.”

(c) Include also an overview of successful applications of these methods to similar problems, such as energy market, optimization of energy production, etc.

Response 2 (c): The paragraph below has been added into the new version of the article (lines 132-135).

“It is worth noting that the DEA and SFA models have been successfully applied in the economic regulation of electricity distribution and transmission utilities, particularly in the definition of the regulatory operational expenditure (OPEX) for each utility, a key element for the annual allowed revenue assessment. [18, 19, 20, 23, 26].”

Point 3: Line 154-155. Please, clarify the suitability of the assumption that the production function is linear in the logarithms, and to which real/practical cases it can be applied.

"Cobb-Douglas" needs suitable biblio references to be well understood.

Response 3: First we replace the word “production” by “cost” at line 171, because equation (2) is a cost function (“Assuming that the production cost function is linear in the logarithms of the variables…”).

To meet the request, we include two new paragraphs in the paper. First in the seccion 2 (lines 153-160):

“It is necessary to recognize that the problem analyzed does not belong to the field of microeconomics. The variables analyzed do not include the variables considered in the Theory of Production and Cost. Thus, the Microeconomic Theory does not present prescriptions about the relationship between non-technical losses (NTL) and their drivers.

It is also necessary to recognize that the non-technical loss is a variable that must be minimized, that is why was adopted the theoretical framework of the cost frontier. However, the estimated frontier function, as proposed in this paper, is not a cost frontier in strictu sensu, it is an efficiency frontier to benchmark NTL”

The second in the seccion 2.1 (lines 178-187):

“In general the SFA models are specified as Cobb-Douglas (CD) or Translog (TL) forms [17, 24]. In the case of cost frontier, the Translog cost function has the most favorable functional properties, it is flexible, but this approach also has problems, it is not parsimonious (there are more parameters to estimate), and this may give rise to econometric difficulties such as multicollinearity and the need for larger samples.

In addition the Translog cost function collapses to a Cobb-Douglas cost function, the latter is a particular case of the former. Cobb-Douglas is less flexible than Translog, but it is parsimonious, i.e. it is the simplest functional form that “gets the job done adequately” [17].

A good example with application of Cobb-Douglas form is the recent comparative study of energy and carbon efficiency for emerging countries using panel stochastic frontier analysis.”.

Note that the model presented in the paper has 7 variables under the Cobb-Douglas specification. The Translog function adds more 28 variables to the model: 7 terms equal to the squares of the logarithms of the variables and more 21 terms, each one equal to the product of the logarithms of two distinct variables.

It is worth mentioning that the main focus of the proposed modelling is the estimation of an efficiency frontier and not the elasticities

The Cobb-Douglas is a well-known function, it models the relationship between the production Y and the inputs: labour L, capital K and raw material M:

Applying a log transformation on equation above we achieve a linear equation on parameter space, i.e., a linear equation on the logarithms of the variables Y, L, K and M.

Finally, as noted by Bhanu Murthy (2002), it is important to remember the following quote from the famous econometrist Lawrence Robert Klein (1974): “Good evidence has been accumulated over the years to suggests that technology on a macroscopic level can pretty well (be) described by the Cobb-Douglas production function ...we recognised that technology could be plausibly explained by this function and that it fitted well with ... a whole system”.

Murthy, K. V. Bhanu, Arguing a Case for Cobb-Douglas Production Function. Review of Commerce Studies, Vol. 20-21, No. 1, January-June, 2002. Available at SSRN: https://ssrn.com/abstract=598082

Klein, L.R. (1974). “Issues in Econometric Studies of Investment Behaviour.” Journal of Economic Literature. Pp.43-50.

Point 4: Sec. 2.1 & 2.2. Uniform notation for logarithm. A notation like log( ) should be preferred, having indicated if the log is natural or decimal (or other).

Response 4: Notation has been uniformed, as suggested, to “log()” in manuscript.

Point 5: Line 201-204. Please, consider the problem of a very small value of gamma that is nearly 0 but not exactly =0: how is it considered? You should specify a threshold T, with gamma <=T and gamma>T, rather than two hard cases gamma=0 and gamma>0. T may be as little as needed to justify your definition of "0".

Moreover, how the fact that gamma is in the interval [0,1] implies that the distribution is a chi-square? Please, include suitable references. And clarifies also the notation where the chi has a subscript "1".

Response 5: Note that we are applying a hypothesis test, a basic procedure for statistical inference. In this case we have the hypothesis H0 gamma=0 and H1 gamma>0. The test described in the paper is a significance test, i.e, it seeks to determine whether gamma is zero or greater than zero.

The threshold T is defined by the level of significance of the test (alpha). In general, alpha is equal to 5%. It is worth mentioning that in hypothesis tests, it is enough to inform the p-value, as indicated in Fig.10. We reject H0 at alpha level of significance if the statistic test exceeds the critical value (the threshold) from chi-square table.

First, note that we are applying maximum likelihood ratio test, whose test statistic is given by LR=-2[ln(Lr)-ln(Lu)] ~ chi-square (J degree of freedom), where ln(Lr) and ln(Lu) are the maximized values of the restricted and unrestricted log-likelihood functions and J is the number of restrictions. (page 224 in Coelli TJ, Rao DSP, O'Donnell CJ, Battese GE. An Introduction to Efficiency and Productivity Analysis. New York: Springer; 2005, reference [17] in this work).

In the case of a half-normal model (the same adopted in this work) the LR statistic is approximated by a chi-square (1 degree of freedom) (page 258 -259 in “Coelli TJ, Rao DSP, O'Donnell CJ, Battese GE. An Introduction to Efficiency and Productivity Analysis. New York: Springer; 2005, reference [17] in this work”).

Then we replace the phrase at line 203 "Since q belongs to the interval [0,1], the distribution of the test statistic can be approximated by a ."

By the following phrase at lines 221-234,

"In the case of half-normal SFA model, the same adopted in this work, the distribution of the test statistic can be approximated by a , i.e., chi-square distribution with 1 degree of freedom [17].

Point 6: Sec. 2.2. It is important that "cross-section" and "panel" terms are well and comprehensively described now that here are compared and contrasted.

Response 6: The following paragraph was included in section 2.2 of the paper (lines 253-260):

“Cross-section data is collected by observing many subjects, such as firms and countries, at the one period of time, for example, the set of annual balance sheets for the last year from each Brazilian distribution utilities. On the other hand, panel data contain observations of multiple variables obtained over multiple periods for the same subjects, for example, the set of annual balance sheets for each Brazilian distribution utility over the last decade.

By incorporating the longitudinal character of the data, making possible to treat any underlying correlation structures in a more appropriate way, studies based on panel data allow for more efficient monitoring of individual units (in this case, utilities) than those based on cross section data.”

Point 7: Line 227. Sentence is quite difficult to understand, using twice "independents" without clarifying what they are: independent variables, maybe? English form must be improved to make this sentence, and other similar ones, readable and understandable.

Response 7 at Lines 265-269: The sentence was adjusted from: “Let uit and vit independents and both independents of the explanatory variables xit, the parameters of the panel data model can be estimated in the same way as the parameters for the cross-section data model.”  To “Let uit and vit independents random variables, both uncorrelated with the explanatory variables xit, the parameters of the panel data model can be estimated in the same way as the parameters for the cross-section data model.”.

Point 8: Line 235-236. Please, clarify what is the meaning of the sentence, in particular how the likelihood test is applied, what "time invariance hypothesis of the efficiency eta =0 means, and where in ref [6] this is demonstrated or clarified. Consider that ref [6] is not sufficient to demonstrate it and needs to be integrated by supplemental references.

Response 8: The SFA models are estimated by maximum likelihood, then in order to assess if the inefficiency is constant across time (time-invariant) or vary in time (time varying) we can fit two SFA models:

Model 1 – SFA model with time-invariant inefficiency, whose maximum log-likelihood is ln(LR)

Model 2 – SFA model with time-varying inefficiency, whose maximum log-likelihood is ln(LU)

The likelihood ratio test statistic (LR statistic) is -2[LR-LU].

The following text were included at the end of line 267:.

“If the inefficiency effect is constant across time then uit = ui and f(t) =1”.

The references [4] and [6], which were wrong, were replaced to the correct references [30] and [17]. The word “invariance” at line 251 was replaced for the correct word is “invariant”.

The paragraph has been replaced by the paragraph below (Lines 273-280):

“Battese and Coelli [30] propose  where h is a parameter to be estimated. In the panel data models, the likelihood ratio test and z test can be applied to evaluate the hypotheses of time invariant efficiency effects H0: h = 0, i.e., the inefficiency effect is constant across time [17, 24]. This specification for f(t) implies that the rank ordering of efficiency scores for firms remains unchanged over time [17]. In addition, in the panel data model estimation the terms uit can be treated as fixed effects or random effects, the latter option being recommended by Kumbhakar [29] and Battese and Coelli [30]”.

Point 9: Sec. 2.1 & Line 274. Ambiguous notation where we do not distinguish the product of theta and PNTL0 from an entire unique variable named "thetaPNTL0". Consider how Ln and ln (that are functions or operators) were used before, without round parentheses but stick to the variables as here at line 274 where you mean really a product, instead.

Response 9: The text has been corrected as indicated below (Lines 318-323):

“Denoting by q (q <1) the efficiency index resulting from the SFA model, the utility's management should reduce from its current value of percentage of non-technical losses (PNTL0) to the target value, i.e., the product q x PNTL0 at the end of a transition period D established by ANEEL, for example, the duration of a four-year tariff review cycle (D = 4). In addition, annual intermediate target values can be established for each year t (1 £ t £ D) based on the geometric rate:”.

Point 10: Fig. 9 and Fig. 11. Add in the figure caption the reason of variability (or in other words the parametrization) of the many curves.

Response 10:

It was inserted a paragraph to better explain Figure (Lines 339-343): “It is shown in Figure 9 the trajectories of the PNTL in the LV networks [9], the variability reflects the huge heterogeneity in the socioeconomic complexity and the different strategies for the non-technical losses management in the Brazilian distribution utilities, for example, the biggest losses are observed in a few companies that serve isolated systems not connected to the distribution network.”

It was inserted a two lines of text to better explain Figure 11 (Lines 371 and 372): “where the line in black is the average and the variability reflects the huge heterogeneity in the current levels of the non-technical losses of the Brazilian distribution utilities.”

Point 11: Results & Discussion. The variables "lowinc","poor","garbage", etc. are only introduced in Results, and they are followed by some interpretation in Discussion: it is in this moment that the reader understands what they refer to. Instead at line 293 when referring to ANEEL the quantities should be defined with some considerations that help clarifying how they will be used, including their range of variation.

Response 11: The "Results" section was divided, part of it being incorporated into a new assignment in the article "Application of the proposed methodology" in order to more clearly separate the explanation about the proposed application and the results. A table (Table 1, Line 356)) was also included with a detail on the explanatory variables used in the application.

However, a broader discussion on explanatory variables is beyond the scope of the paper.

Point 12: Line 343-377. Could you attempt some interpretations and clarification of the results, in particular why some utilities show particular behavior? Eg for utilities at lines 367 and onward, as you have done for utility 5 in lines above.

Response 12: This part of the paper has been revised and expanded to include a more comprehensive explanation of the utilities, which show particular behavior (lines 432-450):

 “The largest deviations (>5%) between the results from models are in the utilities 3, 5, 8, 10, 14, 19, 20, 28 and 38 as indicated by the bottom row of the frequency distribution in Table 4. In this group, the targets defined by the ANEEL model tend to be lower than the targets from SFA approach and they were not achieved in 2018 as indicated in Table 4. In some of these cases, the methodology employed by Aneel requires a reduction in the NTL that is not justifiable for distributors that already have very low NTL levels. For example, the utility 28 registered a very low NTL level in 2018 0.54%), however above the target set by ANEEL (0.24%). The target set by the SFA is 5.36% and therefore avoids an improper penalty of the utility.

In others cases, the methodology used by Aneel does not seem to correctly consider the historical performance of utilities in reducing NTL. For example, utilities 3, 8, 14 and 19. operate in extremely complex areas (high levels of violence and social inequality). Although they continue to register a high level of NTL, these utilities have had a very positive history of NTL reductions over the past few years. In 2018, all these utilities had NTL higher than the Regulatory NTL target set by ANEEL which implied a financial loss for these companies. However, if these companies were compared with the goals established from the SFA, they would all have had NTL levels lower or very close to the target. In recent years, the Agency itself has recognized the limitation of his own model in relation to these utilities and had, in different ways, adjusted the Regulatory Target of these utilities.”

Point 13: To the aim of a better clarification of behavior, and to account for special events (such as the losses of utility 5 occurred in 2018), you should propose a verification based on the data of a different year, such as 2017, in order to derive more general judgments.

Response 13: The objective of this session was to show the feasibility of applying the proposed methodology and its possible benefits and not to carry out an extensive assessment of the impacts of the application of this methodology or analyze the reasons that led to the specific NTL behavior of each of the distributors. Therefore, only one year was chosen (2018). Although, a paragraph was included at the end of this section warning of this study limitation. This paragraph is reproduced below (lines 462-464):

“It is important to note that the objective of this session is only to show the feasibility of applying the proposed methodology and its possible benefits, therefore the selection of only one year (2018). A deeper analysis of the NTL behavior of Brazilian distributors should cover a broader period.”

Round 2

Reviewer 2 Report

Dear Authors, thank you for your kind replies and amendments to the manuscript.